# Targeting PELP1 Attenuates Angiogenesis and Enhances Chemotherapy Efficiency in Colorectal Cancer

**DOI:** 10.3390/cancers14020383

**Published:** 2022-01-13

**Authors:** Jianlin Zhu, Lu Wang, Fan Liu, Jinghua Pan, Zhimeng Yao, Yusheng Lin, Yabing Yang, Xiao Xiong, Kai Li, Yi Yang, Yiran Zhang, Xiaodong Chu, Yunlong Pan, Hao Zhang

**Affiliations:** 1Department of General Surgery, The First Affiliated Hospital of Jinan University, Jinan University, Guangzhou 510632, China; qq123966766@stu2019.jnu.edu (J.Z.); panjh@jnu.edu.cn (J.P.); yangyabing111@stu2020.jnu.edu.cn (Y.Y.); yyflowerst@stu2021.jnu.edu.cn (Y.Y.); susee1992@stu2021.jnu.edu.cn (Y.Z.); stujnu019@stu2021.jnu.edu.cn (X.C.); 2Institute of Precision Cancer Medicine and Pathology, Jinan University Medical College, Guangzhou 510632, China; luella@stu.jnu.edu.cn (L.W.); fanlgodwin@stu2020.jnu.edu.cn (F.L.); yaozhimeng250@jnu.edu.cn (Z.Y.); sunloveflower@stu2021.jnu.edu.cn (Y.L.); xx2015@stu2019.jnu.edu.cn (X.X.); kail59@stu2019.jnu.edu.cn (K.L.); 3Department of Hematology, University Medical Center Groningen, University of Groningen, 9712 CP Groningen, The Netherlands

**Keywords:** colorectal cancer, angiogenesis, PELP1, chemotherapy, tumor vascular normalization

## Abstract

**Simple Summary:**

Excessive angiogenesis is a distinct feature of colorectal cancer (CRC) and plays a pivotal role in tumor development and metastasis. Therefore, it is essential to clarify the underlying mechanism of angiogenesis. In this study, we found that the level of proline-, glutamic acid, and leucine-rich protein 1 (PELP1) was positively correlated with microvessel density (MVD). In vitro and in vivo assays further showed PELP1 regulated angiogenesis via the Signal transducer and activator of transcription 3 (STAT3)/Vascular endothelial growth factor (VEGFA). Notably, we found that inhibition of PELP1 enhanced the efficacy of chemotherapy due to vascular normalization. Thus, targeting of PELP1 may be a potentially therapeutic strategy for CRC.

**Abstract:**

Abnormal angiogenesis is one of the important hallmarks of colorectal cancer as well as other solid tumors. Optimally, anti-angiogenesis therapy could restrain malignant angiogenesis to control tumor expansion. PELP1 is as a scaffolding oncogenic protein in a variety of cancer types, but its involvement in angiogenesis is unknown. In this study, PELP1 was found to be abnormally upregulated and highly coincidental with increased MVD in CRC. Further, treatment with conditioned medium (CM) from PELP1 knockdown CRC cells remarkably arrested the function of human umbilical vein endothelial cells (HUVECs) compared to those treated with CM from wildtype cells. Mechanistically, the STAT3/VEGFA axis was found to mediate PELP1-induced angiogenetic phenotypes of HUVECs. Moreover, suppression of PELP1 reduced tumor growth and angiogenesis in vivo accompanied by inactivation of STAT3/VEGFA pathway. Notably, in vivo, PELP1 suppression could enhance the efficacy of chemotherapy, which is caused by the normalization of vessels. Collectively, our findings provide a preclinical proof of concept that targeting PELP1 to decrease STAT3/VEGFA-mediated angiogenesis and improve responses to chemotherapy due to normalization of vessels. Given the newly defined contribution to angiogenesis of PELP1, targeting PELP1 may be a potentially ideal therapeutic strategy for CRC as well as other solid tumors.

## 1. Introduction

Angiogenesis plays a vital role in the progression and metastasis of solid tumors, including colorectal cancer (CRC) [1,2]. The new vessels not only provide the nutrients to support tumor growth, but also provide a way for the metastasis of tumor [3]. Under normal conditions, the levels of angiogenesis are in a static and stable state due to the balance between pro-angiogenic factors and anti-angiogenic factors [4]. In cancer, this dynamic balance is disrupted, and the tumor microenvironment (TME) secretes excessive pro-angiogenic factors to promote angiogenesis and lead to vessel abnormalization [5]. Previous studies have found that VEGF is the main regulator of tumor angiogenesis [6]. It highly specifically binds to the receptors of vascular endothelial cells to promote tumor angiogenesis. The VEGF family includes VEGF-A, VEGF-B, VEGF-C, VEGF-D, VEGF-E, VEGF-F, and placental growth factor, of which VEGFA is the most important and main component of the VEGF family, and VEGFA has been shown to be associated with the poor prognosis of a variety of solid tumors, including CRC [3]. Therefore, it is considered to be a promising strategy to treat CRC by inhibiting tumor angiogenesis, resulting in tumor death from starvation. At present, a variety of anti-angiogenic drugs targeting VEGF and its receptors have been developed and applied to the clinical treatment of tumors [7]. For example, bevacizumab, which targets VEGF, has been used as a first-line therapy for CRC patients, and has achieved a satisfactory effect [8].

Interestingly, accumulating evidence indicates that some anti-angiogenic drugs potentiate the effects of chemotherapy, immunotherapy, and radiation [9,10,11]. To explain this synergistic phenomenon, the “Tumor vascular normalization” hypothesis was proposed [12]. Indeed, tumor vessels, characterized by abnormal morphology, are highly dysfunctional in their barrier and transport properties [13]. In this hypothesis, anti-angiogenic therapy can induce a temporary reversion of tumor vessels towards a normalized phenotype that includes normalized vessel structure and function [14,15]. However, the tumor vessel normalization effect is transient, reversible, and dose- and time-dependent [16]. Therefore, more theoretical studies on inducing tumor vessel normalization need to be further explored.

Increasing studies have shown that the abnormal expression of a variety of oncogenes in CRC can not only promote tumor proliferation, invasion, and metastasis, but also enhance the secretion of VEGF to affect tumor angiogenesis. For example, B7-H3 promote angiogenesis through activating the NF-κB pathway to enhance VEGFA expression in CRC [17]. TRF2 increases the release and activity of VEGFA by upregulating the expression of SULF2 in CRC [18]. However, the discovery of these angiogenesis-promoting mechanisms has not brought good application prospects.

Proline-, glutamic acid-, and leucine-rich Protein 1 (PELP1) is a scaffold protein, serving as a co-regulatory of various transcription factors and nuclear receptors [19]. PELP1 has the characteristics of proto-oncogenes in tumors, including CRC, and plays an oncogenic function by regulating cell cycle progression, metastasis, hormone therapy resistance, and autophagy [20,21]. In glioma, suppression of PELP1 can down-regulate the expression of VEGFA, but no further studies have been conducted, which suggests that PELP1 has a potential role in regulating tumor angiogenesis [22]. However, the effect of PELP1 on angiogenesis in CRC remains unknown.

In this study, we firstly found that the levels of PELP1 are positively correlated with MVD of clinical CRC tissues. Moreover, PELP1 promotes angiogenesis via STAT3/VEGFA axis in vivo and vitro. Notably, in vivo, we further found that suppression of PELP1 enhances the efficacy of chemotherapy via the normalization of vessels. These results reveal a new role of PELP1 in regulating angiogenesis, and suggest PELP1 may represent an attractive therapeutic target in CRC.

## 2. Materials and Methods

### 2.1. Clinical Patients and Samples

A total of 90 pairs of CRC samples and normal tissues were obtained from CRC patients who underwent surgeries at the First Affiliated Hospital of Jinan University (Guangzhou, China).All samples were histopathologically and clinically diagnosed as CRC. Patients who underwent preoperative neoadjuvant chemotherapy or radiotherapy for CRC were excluded from this study.

### 2.2. Conditioned Medium (CM)

CRC cells (1.0 × 10^6^) were cultured in 6-well plates for overnight and then the culture medium was changed to 2 mL basal DMEM medium in each well. After 24 h, the conditioned medium was collected and centrifuged (1500× *g*, 15 min) and stored at −80 °C. The CM was used for ELISA, wound healing, transwell, and tube formation assays.

### 2.3. Immunofluorescence

Immunofluorescence was performed as described previously [23]. Briefly, the cells or sections were processed before being subjected to immunofluorescence analysis and then the primary antibody against PELP1, human CD31 (Cat.No.AF3628, R&D, Minneapolis, MN, USA), CD34 (Cat. No.35493, SAB), VEGFA, and Ki-67 (Cat.No.48871, SAB, Miami Beach, FL, USA) were incubated with specific primary antibodies overnight at 4 °C, followed by Alexa Fluor 594 (Red)-conjugatedsecondary antibody or 488 (Green)-conjugated secondary antibody for 1 h in darkness. Nuclei were stained with DAPI. All images were captured using Bio-Tek/Cytation5 system and analyzed by Image J software (version 1.46; National Institutes of Health).

### 2.4. Woundhealing Assay

To test the migration capacity of HUVECs cultivated with CM, each well of a 6-well plate was seed with 5 × 10^5^ HUVECs for the wound healing assay. After 100% confluence of cells were achieved, a micropipettor (200 µL) was used to generate a scratch on the bottom of the 6-well plate, then the cells were washed twice with PBS, and the medium was changed to CM without or with recombinant human VEGF (rVEGFA) (Cat.No. 100-20, Peprotech, London, UK). All cells were cultivated with CM and observed at 0, 12, 24, and 48 h post scrape. Microscopy was used to observe and photograph cell migration to the scratch area and calculate the healing area of the wound. All the images we obtained were analyzed by Image J software (version 1.46; National Institutes of Health).

### 2.5. Transwell Invasion Assays

Transwell was performed as described previously [24]. HUVECs invasion assays were performed by transwell chambers (Cat.No.353097, Corning, New York, NY, USA) coated with Matrigel matrix (Cat.No. 354248, Corning) 200 μL of basal ECM containing 1.0 × 10^4^ HUVECs was added to the upper chamber, and 800 μL of CM with or without rVEGFA was added to the lower chamber and incubated for 36 h at 37 °C with 5% CO^2^. Prior to fixation, cells on the upper membrane were removed with cotton swabs. The cells on the membranes were fixed and stained using 0.1% crystal violet for 10min at room temperature. Images were captured using light microscopy and invasion cells were counted using Image J software (version 1.46; National Institutes of Health) in three randomly chosen fields per well.

### 2.6. Tube Formation Assay

HUVECs were seeded in 96-well plates coated with 50 μL Matrigel matrix at a density of 1.0 × 104 cells/well and cultured in CM for 4 h at 37 °C with 5% CO^2^. Four images per well were randomly captured by microscopy and evaluated by Image J software (version 1.46; National Institutes of Health).

## 3. Results

### 3.1. PELP1 Is Highly Expressed in Clinical CRC Tissue Samples and Positively Correlated with MVD

To systematically explore the role of PELP1 in angiogenesis of CRC, we first conducted gene set enrichment analysis (GSEA) on the dataset GSE29263, GSE65979 and revealed that the angiogenesis signatures were highly correlated with PELP1 expression (Appendix A). We then conducted immunofluorescence analyses to compare the expression levels of PELP1 and CD31/CD34 (human endothelial marker for MVD) in 90 clinical CRC tissues with their paired normal tissues (Figure 1A). The result showed that PELP1 and CD31 expressions were significantly higher in the CRC tissues (Figure 1B,C). Moreover, we divided 90 patients into two groups, namely, high-PELP1 group (above median value) and low-PELP1 group (below median value), and found that high-PELP1 group had significantly higher CD31 expression than the low-PELP1 group (Figure 1D). More importantly, further analysis of these data showed that the expression level of PELP1 in CRC tissue samples was positively correlated with the level of CD31 (Figure 1E). In addition, we selected CD34 as another marker of endothelial cells and repeated the above experimental steps. We found that the results are consistent with the previous ones (Figure 1F–I). Overall, these results suggested that the upregulation of PELP1 is closely correlated with angiogenesis, and PELP1 plays a pivotal role in CRC angiogenesis.

### 3.2. Down-Regulation of PELP1 Attenuates Angiogenesis In Vitro

We initially conducted RT-qPCR and immunoblot analysis and found that PELP1 expression level was higher in the CRC cell lines than in the immortalized colonic epithelium (Appendix A).To validate the effect of PELP1 on angiogenesis in vitro, we choose two cell lines HCT116 and HT29 with moderate levels of PELP1 to examine the effect of PELP1 on the angiogenesis makers (Appendix A). CCK8 assay showing knockdown of PELP1 inhibited cell proliferation of HCT116 and HT29 cells, which is consistent with previous reports (Appendix A). Angiogenesis is a complex process, including the proliferation, migration, and differentiation of vascular endothelial cells [25]. HUVECs were used to perform wound healing, transwell and tube formation assays to assess its migration, invasion, and tube formation capabilities to evaluate the effect of PELP1 on angiogenesis in this study (Figure 2A). We found that knockdown PELP1 (shPELP1) in CRC cells significantly decreased the migration, invasion, and tube formation of HUVECs compared to the control (shCtrl) (Figure 2B–D). On the other hand, overexpression of PELP1 enhanced these capacities of HUVECs (Appendix A). These results support the notion that PELP1 may be involved in the regulation of CRC cell angiogenesis.

### 3.3. Downregulation of PELP1 Attenuates Angiogenesis by Reducing VEGFA Expression

Tumor angiogenesis is regulated by various cytokines and growth factors in TME [26]. In order to understand the mechanism in PELP1-regulated CRC angiogenesis, weutilized RT-qPCR to detect the mRNA expressions of multiple angiogenesis-related cytokines in PELP1-modified CRC sublines cells. Interestingly, the mRNA expressions of VEGFA, Matrix metalloprotein-2 (MMP2), and Platelet derived growth factor-BB (PDGF-BB) were significantly down- and up-regulated by PELP1 depletion and overexpression, respectively (Figure 3A; Appendix A). Given the pivotal roles of VEGFA and PDGF-BB in tumor angiogenesis, we conducted ELISA assay on these two genes and found that only VEGFA is significantly down- and up-regulated by PELP1 depletion and overexpression, respectively (Figure 3B; Appendix A). In addition, immunoblot assays also confirmed that VEGFA was down- and up-regulated by PELP1 depletion and overexpression in CRC cells (Figure 3C; Appendix A). Similarly, immunofluorescence assay showed that VEGFA is significantly down-regulated by PELP1 depletion and up-regulated by PELP1 overexpression (Figure 3D; Appendix A). Moreover, we further verified the relationship between PELP1 and VEGFA expression levels in clinical tissues samples by immunofluorescence (Figure 3E). We found that VEGFA expressions were significantly higher in the CRC tissues (Figure 3F). Further analysis showed thathigh-PELP1 group had significantly higher VEGFA expression than the low-PELP1 group (Figure 3G) and the expression level of PELP1was positively correlated with the expression of VEGFA (Figure 3H). In addition, in order to clarify PELP1-induced angiogenesis via VEGFA, we added human recombinant VEGFA (rVEGFA) into the CM from PELP1-depleted cells, and found that the inhibitory effect on HUVECs was reversed (Figure 3I–K; Appendix A). More importantly, we established PELP1-overexpressing cells with knockdown of VEGFA by specific siRNA and verified it by RT-qPCR, immunoblot and ELISA (Appendix A). The results showed that the effect of PELP1 on angiogenesis was attenuated by VEGFA siRNA (Appendix A). Taken together, these data provide strong evidence that PELP1 induced angiogenesis through VEGFA.

### 3.4. PELP1 Promotes Angiogenesis via the STAT3/VEGFA Axis

Tumor-derived angiogenic factors were regulated by various signaling pathways, such as STAT3, ERK, AKT, and NF-κB pathways [17,27,28,29]. Previous reports have shown that PELP1 regulate the phosphorylation of STAT3 (p-STAT3), and STAT3 regulate the expression of VEGFA to affect the angiogenesis of tumor [30,31]. To determine if STAT3 was involved in PELP1-mediated VEGFA upregulation, the results showed that knockdown PELP1 in CRC cells obviously reduced p-STAT3 without affecting the total level of STAT3 (Figure 4A). Furthermore, Stattic, an inhibitor that can specifically inhibit the phosphorylation of STAT3, was used in this study. CRC cells with PELP1 overexpressed were incubated with or without Stattic (5 µM, 24 h), and it was found that overexpression PELP1 in CRC cells significantly reduced the expression of VEGFA with Stattic (Figure 4B). Immunoblot analysis showed that Stattic appeared to counteract PELP1-enhanced expressions of p-STAT3 and VEGFA (Figure 4C). Moreover, ELISA and immunofluorescence assays also showed that Stattic neutralized the effect of PELP1 (Figure 4D,E; Appendix A). In addition, to further confirm that PELP1 promotes angiogenesis through STAT3, we collected CM from CRC cells with PELP1 overexpressed with or without Stattic treatment to test the capacity of HUVECs. As we expected, when the PELP1-overexpressing cells were treated with Stattic, it significantly reduced the capacity of HUVECs (Figure 4F–H; Appendix A). Collectively, these data suggest that PELP1 promoted angiogenesis through STAT3/VEGFA axis in CRC.

### 3.5. Suppressing PELP1 Attenuates Angiogenesis via STAT3/VEGFA Axis In Vivo

To investigate the role of PELP1 on tumor angiogenesis in vivo, we inoculated HCT116-shCtrl and HCT116-shPELP1 cells into flanks of nude mice. The animals were sacrificed at the end of the experiment, and the tumors were dissected and weighed. The tumors derived from the cells with PELP1-depleted tumors were smaller and lighter (Figure 5A–C). Of note, immunoblot and immunofluorescence assay showed that knockdown PELP1 suppressed the STAT3/VEGFA axis (Figure 5D,E). Furthermore, to verify the effect of PELP1 on tumor angiogenesis, the results showed that suppression of PELP1 significantly reduced MVD (Figure 5F,G). These data altogether indicate that PELP1 suppression reduced angiogenesis via the STAT3/VEGFA axis.

### 3.6. PELP1 Suppression Enhances the Effectiveness of Chemotherapy via Vascular Normalization

Increasing clinical evidences show that anti-angiogenesis therapy enhances the efficacy of chemotherapy in solid tumors [32,33]. In this study, we found that PELP1 suppression can inhibit angiogenesis in CRC. We further explore whether inhibition of PELP1 combined with chemotherapy can enhance the efficacy of chemotherapy. As shown in Figure 6A–D, administration of the chemotherapy drug to mice xenografted with PELP1 knockdown cells enhanced the antitumor efficacy. Meanwhile, immunofluorescence staining of Ki67 showed that cisplatin (Cpt) treatment significantly reduced the proliferation in PELP1 knockdown group (Figure 6E). The above results indicated that suppression of PELP1 enhances the efficacy of chemotherapy, and we further explore its mechanism. The efficacy of chemotherapy depends on drug delivery in the tumor tissue [34]. It is reported that anti-angiogenesis therapy enhances the efficacy of chemotherapy due to the normalization of tumor vessels [35]. Given the results in this study showed that PELP1 depletion reduced VEGFA to decrease angiogenesis in CRC. Therefore, we hypothesized that PELP1 suppression reduced the secretion of VEGFA to induce tumor vessel normalization. To confirm this hypothesis, double immunofluorescent staining of CD31 and α-SMA (pericyte marker) in tumor xenograft tissues revealed that PELP1 depletion significantly increased the pericytes coverage (Figure 6F). Moreover, tumor vascular normalization can reduce tissue hypoxia [36].To detect the hypoxic areas in tumor tissues, pimonidazole (PIMO) staining revealed that the hypoxic areas were significantly decreased in PELP1 knockdown tissues (Figure 6G). In addition, HPLC assay revealed that the accumulation of Cpt is increased in PELP1 knockdown tissues (Figure 6H). Taken together, these data suggest that PELP1 suppression could induce vascular normalization and then enhance chemotherapy efficiency by improving delivery of anti-tumor drugs into tumor tissues.

## 4. Discussion

Abnormal angiogenesis is an important feature of CRC. The present study revealed that the high expression of PELP1 is closely related with angiogenesis in CRC.We proved that PELP1 promotes tumor angiogenesis by activating the STAT3 pathway. It is worth noting that suppression of PELP1 reduces VEGF expression and secretion, and then enhances the efficacy of tumor chemotherapy via vascular normalization, which provides a novel target for anti-angiogenesis in CRC.

Previous studies have shown that PELP1 is highly expressed in a variety of tumors, such as breast cancer, ovarian cancer, brain cancer, lung cancer, and colorectal cancer, and promotes tumor growth by multiple signaling pathways [22,36,37,38,39,40]. In this study, our results indicated that PELP1 was up-regulated in CRC tissues compared with normal colorectal specimens, and that inhibition of PELP1 led to inhibition of tumor growth. These results were consistent with those observed in other cancers. It has been shown that PELP1 promotes tumor metastasis, but there is no report on the relationship between PELP1 and tumor vessels. Tumor vessels not only support tumor growth, but also provide a way for tumor metastasis [3]. Previous studies reported that MVD was positively correlated with tumor lymph node metastasis, and distant metastasis in a variety of tumors [41]. In this study, we found for the first time that PELP1 was positively correlated with MVD in CRC and proved that PELP1 regulated tumor angiogenesis through the STAT3/VEGFA axis. Most of the previous studies showed that PELP1 promoted tumor metastasis by enhancing tumor invasion and migration ability. Our results reveal the specific function of PELP1 in promoting angiogenesis, thus to facilitate tumor metastasis.

Tumor angiogenesis is a complex multi-step process, which is regulated by cytokines and growth factors secreted from different types of cells in the tumor microenvironment, including tumor itself [26]. To clarify how PELP1 regulates angiogenesis in this study, we analyzed the mRNA expression levels of angiogenesis-related factors and found that the mRNA expression level of VEGFA and PDGF-BB is positively correlated with PELP1 mRNA expression. Furthermore, ELISA was used to analysis the protein expression of VEGFA and PDGF-BB in the CM from CRC cells expressing shPELP1 or PELP1. The results showed that PELP1 can regulate the secretion of VEGFA, but has no effect on PDGF-BB. The results from immunoblot and Immunofluorescence further showed that in PELP1 knockdown CRC cells, the protein expression level of VEGFA was significantly decreased. Moreover, in 90 CRC tissue samples, we found that the expression of VEGFA was significantly higher in tumor than that in paired normal tissues. In tumor tissue, expression of VEGFA was positively correlated with those ofPELP1. In addition, the rescue experiments revealed that rVEGFA treatment abolished the inhibitory effects of CM from shPELP1 CRC cells on HUVECs function, while VEGFA siRNA reversed the effects of CM from PELP1 CRC cells on HUVECs. These results clearly indicated that PELP1 regulated tumor angiogenesis through VEGFA. In this study, we found a novel mechanism that PELP1 regulates tumor angiogenesis by promoting the expression of VEGFA, and may become a new target for anti-tumor angiogenesis therapy.

In the actual application of anti-angiogenesis drugs, we found that anti-angiogenesis therapy does not improve survival and may increase the risk of tumor metastasis [42,43]. The theory of tumor vascular normalization provides a new basis for expanding the application of anti-angiogenesis drugs. At present, there are various methods for inducing tumor vascular normalization, such as humanized monoclonal antibody, tyrosine kinase inhibitor, and special reagents: nanoparticles, thalidomide, and metformin [44,45,46,47,48,49]. In this study, we found that tumor vessels normalization can be induced by inhibition of PELP1 in CRC. Of note, PELP1 suppression combined with chemotherapeuticdrug has a synergistic anti-tumor effect. It has been reported that targeting PELP1 can induce tumor vessels normalization through attenuating angiogenesis and enhance the efficacy of chemotherapy via increasing the delivery of chemotherapeutic drug into tumor tissue (Figure 7).Currently, the main problem of tumor vascular normalization is that the window period is short, and the window period is difficult to predict [50]. These issues were not further explored in this study. Many reports have shown that the normalization of tumor vessels improve the efficacy of radiotherapy and immunotherapy, whether there is the same effect on the normalization of vessels induced by PELP1 is not clear in this study [51,52]. We will explore the above issues in follow-up studies. Above all, these findings provide a new insight in the clinical application of the mechanisms of various oncogenes regulating angiogenesis.

## 5. Conclusions

In conclusion, our results reveal a new mechanism that PELP1 promotes angiogenesis by STAT3/VEGFA axis, and suppression of PELP1 enhances the effectiveness of chemotherapy via vascular normalization. Therefore, targeting PELP1 combining with chemotherapeutic drug may be a promising strategy for treatment of CRC.

## Figures and Tables

**Figure 1 cancers-14-00383-f001:**
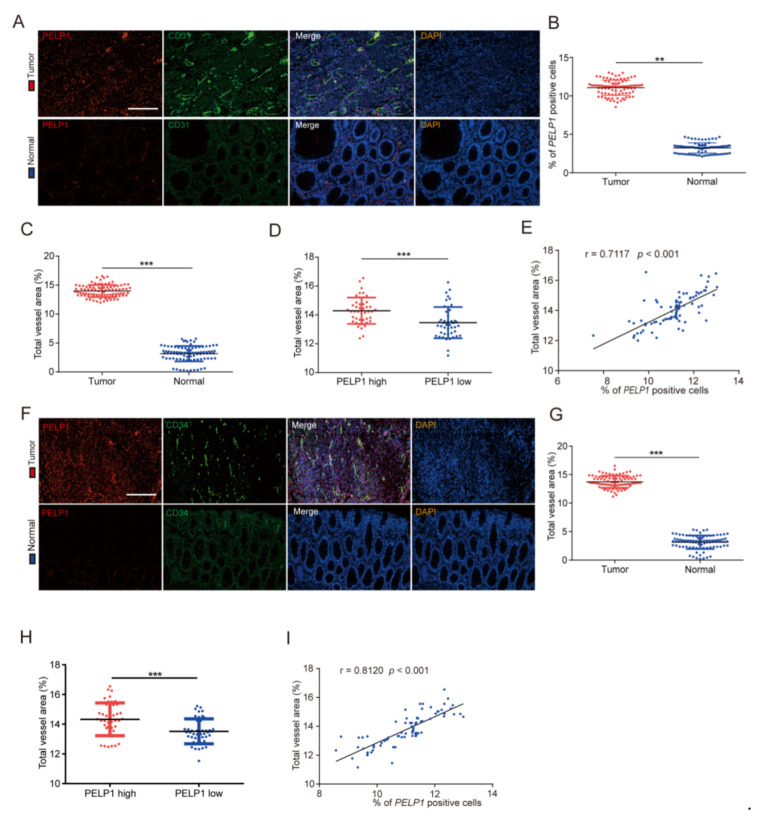
Correlation analysis between PELP1 and MVD in clinical CRC samples: (**A**) Representative images of immunofluorescence for PELP1 and CD31 in 90 CRC tissues and their paired normal tissues. Scale bar: 200 µm. (**B**) Quantification of PELP1-positive cells in CRC tissues and paired normal tissues. (**C**) Quantification of MVD (CD31) in CRC tissues and paired normal tissues. (**D**) Quantification of MVD (CD31) in PELP1 low and PELP1 high CRC tissues. (**E**) Correlation analysis between PELP1 and MVD (CD31) in CRC tissues. (**F**) Representative images of immunofluorescence for PELP1 and CD34 in 90 CRC tissues and their paired normal tissues. Scale bar: 200 µm. (**G**) Quantification of MVD (CD34) in PELP1 low and PELP1 high CRC tissues. (**H**) Quantification of MVD (CD34) in PELP1 low and PELP1 high CRC tissues. (**I**) Correlation analysis between PELP1 and MVD (CD34) in CRC tissues. Data are shown as the means of three independent experiments or representative data. Error bars indicate SD. ** *p* < 0.01, *** *p* < 0.001, by Student’s *t*-test.

**Figure 2 cancers-14-00383-f002:**
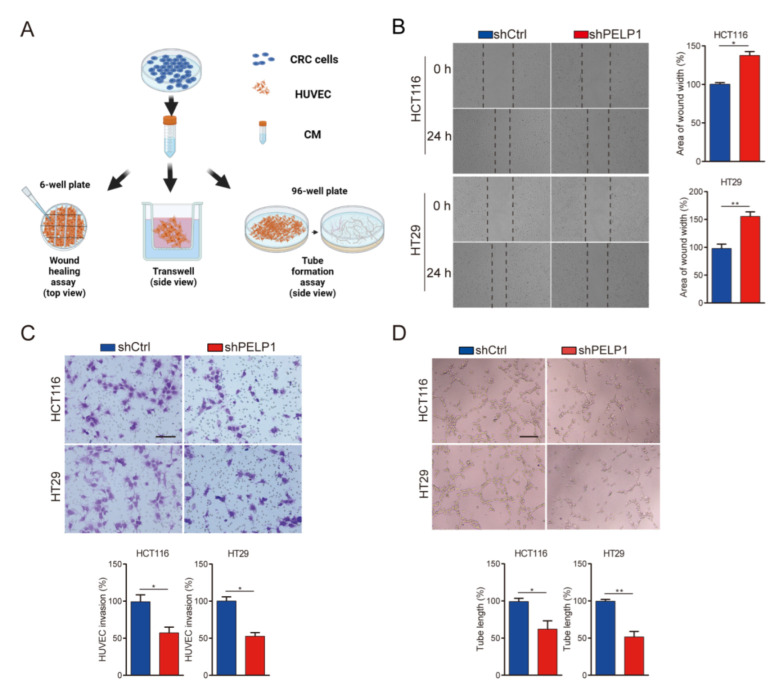
Down-regulation of PELP1 attenuates angiogenesis in vitro: (**A**) Schematic diagram of the HUVECs experiments in this study. (**B**) Representative images of wound healing in HUVECs treated with CM from shCtrl CRC cells or shPELP1 CRC cells (left panel). Scale bar: 200 µm. Histograms with the fold change in wound closure formed by the indicated cells (right panel). (**C**) Representative images of cells invasion in HUVECs treated with CM from shCtrl CRC cells or shPELP1 CRC cells (upper panel). Scale bar: 200 µm. Histograms with the fold change in the number of invasion cells formed by the indicated cells (lower panel). (**D**) Representative images of tube formation in HUVECs treated with CM from shCtrl CRC cells or shPELP1 CRC cells (upper panel). Scale bar: 200 µm. Histograms with the fold change in the length of tube-like formation formed by the indicated cells (lower panel). Data are shown as the means of three independent experiments or representative data. Error bars indicate SD.* *p* < 0.05, ** *p* < 0.01, by Student’s *t*-test.

**Figure 3 cancers-14-00383-f003:**
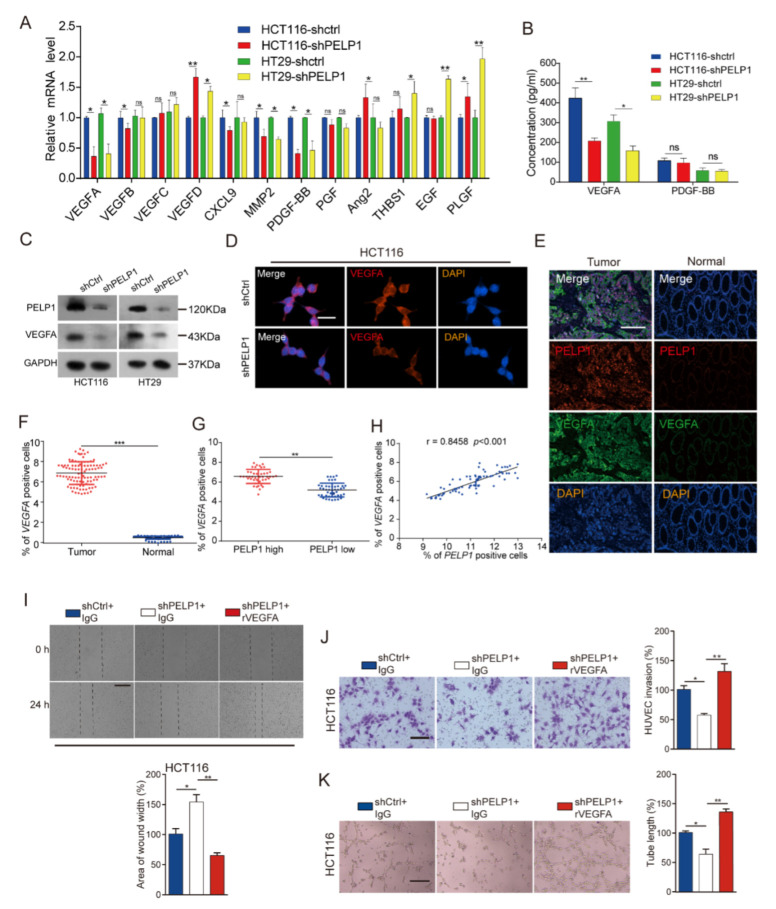
Down-regulation of PELP1 attenuates angiogenesis by reducing the expression of VEGFA: (**A**) The mRNA levels of the 12 angiogenesis-related genes in shCtrl cells or shPELP1 cells. GAPDH was used as an internal control. (**B**) ELISA analysis of VEGFA and PDGF-BB expression in shCtrl cells or shPELP1 cells. (**C**) Immunoblot analysis of PELP1 and VEGFA expressions in shCtrl cells or shPELP1 cells. GAPDH was used as an internal control. (**D**) Representative images of immunofluorescence for VEGFA in HCT116-shCtrl CRC cells or HCT116-shPELP1 CRC cells. Scale bar: 30 µm. (**E**) Representative images of immunofluorescence for PELP1 and VEGFA in 90 CRC tissues and their paired normal tissues. Scale bar: 200 µm. (**F**) Quantification of VEGFA-positive cells in CRC tissues and paired normal tissues. (**G**) Quantification of VEGFA-positive cells in PELP1 low and PELP1 high CRC tissues. (**H**) Correlation analysis between PELP1 and VEGFA in CRC tissues. (**I**) Representative images of wound healing in HUVECs treated with CM from HCT116-shCtrl CRC cells or HCT116-shPELP1 CRC cells treated with or without rVEGFA and IgG (upper panel). Scale bar: 200 µm. Histograms with the fold change in wound closure formed by the indicated cells (lower panel). (**J**) Representative images of cells migration in HUVECs treated with CM from HCT116-shCtrl CRC cells or HCT116-shPELP1 CRC cells treated with or without rVEGFA and IgG (left panel). Scale bar: 200 µm. Histograms with the fold change in the number of invasion cells formed by the indicated cells (right panel). (**K**) Representative images of tube formation in HUVECs treated with CM from HCT116-shCtrl CRC cells or HCT116-shPELP1 CRC cells treated with or without rVEGFA and IgG (left panel). Scale bar: 200 µm. Histograms with the fold change in the length of tube-like formation formed by the indicated cells (right panel). Data are shown as the means of three independent experiments or representative data. Error bars indicate SD. ns: no significance, * *p* < 0.05, ** *p* < 0.01, *** *p* < 0.001 by Student’s *t*-test.

**Figure 4 cancers-14-00383-f004:**
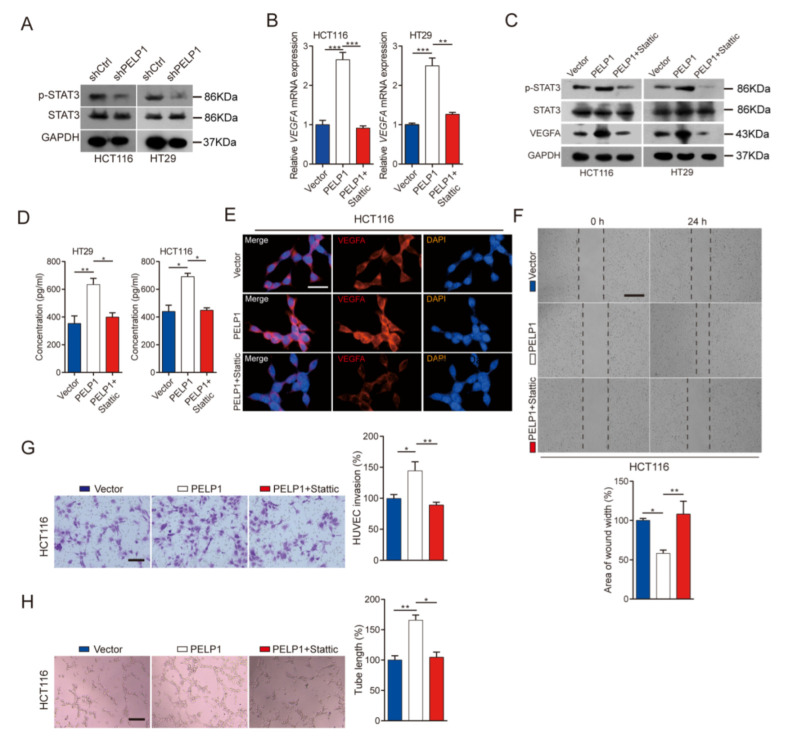
PELP1 promotes VEGFA-mediated angiogenesis via the STAT3: (**A**) Immunoblot analysis of p-STAT3 and STAT3 expressions in shCtrl cells or shPELP1 cells. GAPDH was used as an internal control. (**B**) RT-qPCR analysis of VEGFA in PELP1 overexpressed cells treated with or without Stattic. GAPDH was used as an internal control. (**C**) Immunoblot analysis of p-STAT3, STAT3, and VEGFA expressions in PELP1 overexpressed cells treated with or without Stattic. (**D**) ELISA analysis of VEGFA expression in PELP1 overexpressed cells treated with or without Stattic. (**E**) Representative images of immunofluorescence for VEGFA in HCT116-PELP1 CRC cells treated with or without Stattic. Scale bar: 30 µm. (**F**) Representative images of wound healing in HUVECs treated with CM from HCT116-PELP1 CRC cells treated with or without Stattic (upper panel). Scale bar: 200 µm. Histograms with the fold change in wound closure formed by the indicated cells (lower panel). (**G**) Representative images of cell invasion in HUVECs treated with CM from HCT116-PELP1 CRC cells treated with or without Stattic (left panel). Scale bar: 200 µm. Histograms with the fold change in the number of invaded cells formed by the indicated cells (right panel). (**H**) Representative images of tube formation in HUVECs treated with CM from HCT116-PELP1 CRC cells treated with or without Stattic (left panel). Scale bar: 200 µm. Histograms with the fold change in the length of tube-like formation formed by the indicated cells (right panel). Data are shown as the means of three independent experiments or representative data. Error bars indicate SD.* *p* < 0.05, ** *p* < 0.01, *** *p* < 0.001 by Student’s *t*-test.

**Figure 5 cancers-14-00383-f005:**
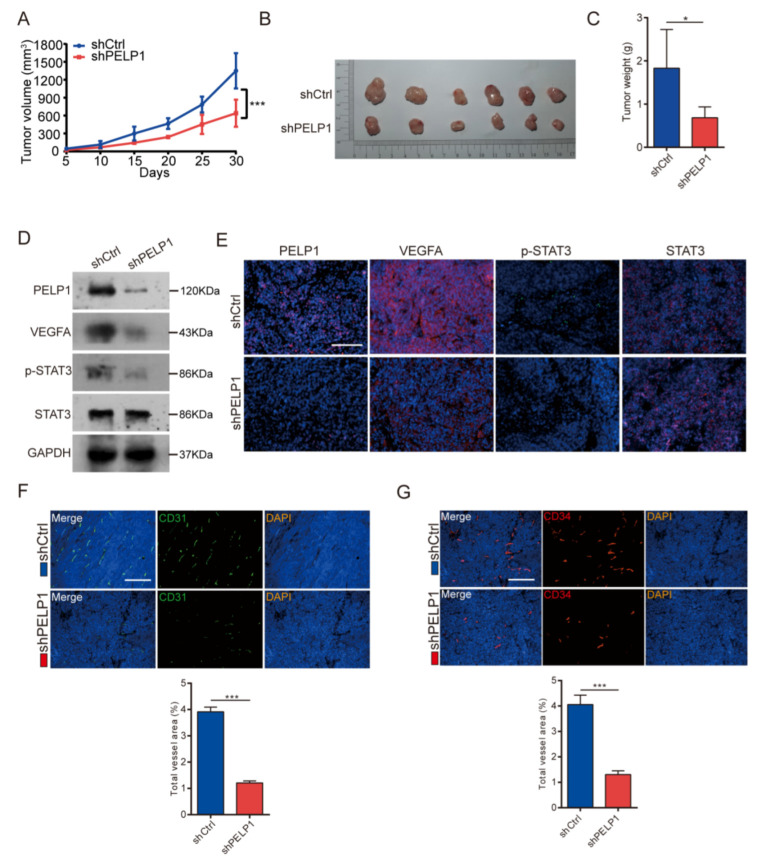
PELP1 suppression attenuates angiogenesis via STAT3/VEGFA axis in vivo: (**A**) Tumor growth curves showed the measure tumor volumes over time. (**B**,**C**) Representative tumors (**B**) and the bar charts of the weights (**C**) of all tumors harvested at the end of the experiments were shown. (**D**,**E**) Expression of PELP1, VEGFA, p-STAT3, and STAT3 detected by immunoblotting (**D**) and Immunofluorescent (**E**) in tumors derived from nude mice injected with HCT116 cells stably expressing shPELP1 or shCtrl. GAPDH was used as an internal control. Scale bar: 200 µm. (**F**,**G**) Representative images of immunofluorescence for MVD (CD31 and CD34) in tumors derived from nude mice injected with HCT116 cells stably expressing shPELP1 or shCtrl (upper panel). Scale bar: 200 µm. Quantification of MVD in the harvested tumors (lower panel). Data are shown as the means of three independent experiments or representative data. Error bars indicate SD.* *p* < 0.05, *** *p* < 0.001 by Student’s *t*-test.

**Figure 6 cancers-14-00383-f006:**
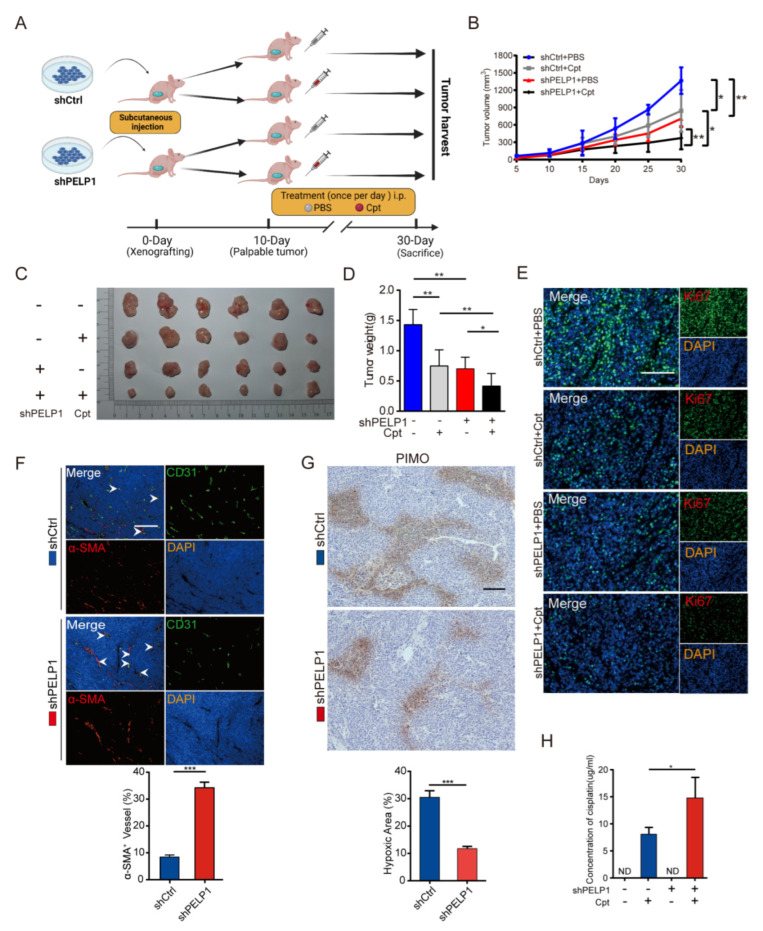
PELP1 suppression enhance efficacy of chemotherapy via vascular normalization: (**A**) Scheme indicating the timing of xenografting and longitudinal treatment. (**B**–**D**) Growth curves (**B**), the harvest tumors (**C**) and the weight of tumors (**D**) derived from nude mice injected with HCT116 cells stably expressing shPELP1 or shCtrl and with or without Cpt. (**E**) Representative images of immunofluorescence for Ki67 in tumor derived from nude mice injected with HCT116 cells stably expressing shPELP1 or shCtrl and with or without Cpt. Scale bar: 100 µm. (**F**) Double immunostaining (upper) for CD31 (Green) and α-SMA (Red) in sections of tumors derived from nude mice injected with HCT116 cells stably expressing shPELP1 or shCtrl, and quantification (lower) of percentage of α-SMA+/CD31+ vessels to assess pericyte coverage. Scale bar: 200 µm. (**G**) Representative image (upper panel) and quantification (lower panel) of PIMO staining in tumors derived from nude mice injected with HCT116 cells stably expressing shPELP1 or shCtrl. Scale bar: 100 µm. (**H**) HPLC analysis of Cpt concentration in tumor derived from nude mice. Data are shown as the means of three independent experiments or representative data. Error bars indicate SD.* *p* < 0.05, ** *p* < 0.01, *** *p* < 0.001 by Student’s *t*-test.

**Figure 7 cancers-14-00383-f007:**
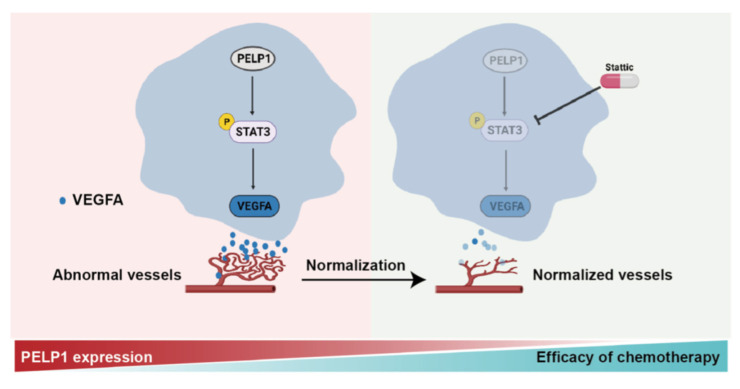
PELP1 regulates the expression of VEGFA by the STAT3 to promote tumor angiogenesis in CRC. Further, PELP1 suppression could enhance the efficacy of chemotherapy, which is caused by the normalization of vessels.

## Data Availability

The data presented in this study will be made available upon reasonable request.

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
