# Peer review of "Targeting PELP1 Attenuates Angiogenesis and Enhances Chemotherapy Efficiency in Colorectal Cancer"

_cancers, 2022, doi:10.3390/cancers14020383_

Round 1

Reviewer 1 Report

Angiogenesis plays an important role in the cancer progression and chemo-resistance. The current study has focused on the role of PELP1 on angiogenesis in colorectal cancer. The authors have performed various techniques to confirm their hypothesis and the results are coherent and reliable. Current manuscript will be of interest for many readers and deserves to be published in Cancers. However, some changes in main text are needed to improve quality of articles.

  1. PELP1 depletion reduced tumor growth in xenograft model. this reduction may be due to cancer cell proliferation in vitro as well as angiogenesis. The authors should investigate the effect on the proliferation of cancer cells in vitro.
  2. Conclusion section can be elaborated and improved by adding more statements about limitations of current work and providing more directions for future studies.
  3. Some English correction is required (ex. Line 229)

Author Response

We would like to express our sincere appreciation to the reviewers for their insightful evaluation and valuable critique of our manuscript. Based on this critique, we have created a stronger, more focused manuscript. Please see the attachment

Reviewer 2 Report

Considering the extensive research and the contribution of the authors to the progress in cancer research with enhanced treatment potential, through deciphering a mechanistic pathway with clinical cases complemented with cell lines, the clarity in data presentation and appropriate Figures and Tables in addition to the originality of research, the quality of this paper deserves appreciation and merits my support.

The present work is significant from the point of view of clinical management of cancers. It opens doors to test this aspect in other cancers besides CRC, with expected positive results. Additionally, the mechanism can be extrapolated to other diseases with symptoms of higher MVD.

There are a few minor points that need to be checked:

  1. Some abbreviations used in the paper require expanding on first use, such as shPELP1, shCrtl PDGF-BB.
  2. The figure legends must explain data points and cell lines used to benefit a wide readership who are not cancer biologists. For eg., Figure 2A has many cell lines tested. It will be better to write that FHC is immortalized epithelial cell.
  3. Figure 2 can be divided into the expression of PELP1 as one figure and HUVEC based assays as another figure to increase clarity and link the two experiments as sequential analysis.
  4. Figure 3 F,G, and H must mention on the y-axis that the + cells were positive for VEFG. It is a bit confusing otherwise.
  5. Minor grammar and spell check are needed. For eg. Line 76shoudl be ... activating the NFkB …, leucine is spelled as geucine (line 80), line 98 … patients who underwent instead of patients underwent…
  6. Line 319. The citation (32) is a study on the stromal cells that pose resistance to antiangiogenesis drugs and doesn’t qualify as supporting evidence to enhanced efficacy of chemotherapy …. Kindly check. I would suggest:  Self-anti-angiogenesis nanoparticles enhance anti-metastatic-tumor efficacy of chemotherapeutics, 2021: Recent advances in, and challenges of, anti-angiogenesis agents for tumor chemotherapy based on vascular normalization, 2021, State-of-the-Art in Antiangiogenic Agents in Cancer Therapy, 2021

Author Response

We would like to express our sincere appreciation to the reviewers for their insightful evaluation and valuable critique of our manuscript. Based on this critique, we have created a stronger, more focused manuscript. Please see the attachment.
